# Simulated single-layer forest canopies delay Northern Hemisphere snowmelt

**Markus Todt[1], Nick Rutter[1], Christopher G. Fletcher[2], and Leanne M. Wake[1]**

[1]Department of Geography, Northumbria University, Newcastle upon Tyne, UK

[2]Department of Geography and Environmental Management, University of Waterloo, Waterloo, Ontario, Canada

**Correspondence:** Markus Todt (markus.todt@northumbria.ac.uk)

**Abstract.**

Single-layer vegetation schemes in modern land surface models have been found to overestimate diurnal cycles in longwave radiation beneath forest canopies. This study introduces an empirical correction, based on forest stand-scale simulations, which reduces diurnal cycles of sub-canopy longwave radiation. The correction is subsequently implemented in land-only simulations of the Community Land Model version 4.5 (CLM4.5) in order to assess the impact on snow cover. Nighttime underestimations of sub-canopy longwave radiation outweigh daytime overestimations, which leads to underestimated averages over the snow cover season. As a result, snow temperatures are underestimated and snowmelt is delayed in CLM4.5 across evergreen boreal forests. Comparison with global observations confirms this delay and its reduction by correction of sub-canopy longwave radiation. Increasing insolation and day length change the impact of overestimated diurnal cycles on daily average sub-canopy longwave radiation throughout the snowmelt season. Consequently, delay of snowmelt in land-only simulations is more substantial where snowmelt occurs early.

## 1 Introduction

Forest canopy cover modulates longwave radiation received by the ground, which consequently differs from atmospheric longwave radiation. This process is called longwave enhancement and has been shown to result in substantial positive net longwave radiation of the surface when snow cover is prevalent, especially under clear skies and during snowmelt (Webster et al., 2016). In contrast, net longwave radiation fluxes are typically negative for snow under clear-sky conditions in unforested areas as has been observed for ever-green Canadian boreal forests (Harding and Pomeroy, 1996; Ellis et al., 2010). Moreover, forest cover has been reported to enhance snowmelt for a subarctic open woodland during overcast days and early in the snowmelt season (Woo and Giesbrecht, 2000). However, the impact of forest coverage on snowmelt varies regionally as a function of forest density and meteorological conditions, with the importance of shortwave and longwave radiation changing throughout the snowmelt season (Sicart et al., 2004; Lundquist et al., 2013).

Meteorological conditions control longwave enhancement as clear skies increase insolation and thereby vegetation temperature while radiative temperature of the sky is reduced (Sicart et al., 2004; Lundquist et al., 2013; Todt et al., 2018). Therefore, values for longwave enhancement, i.e. the ratio of below-canopy to above-canopy longwave radiation, are higher under clear skies but close to 1 or even smaller for overcast conditions due to similar vegetation temperature and radiative temperature of the sky. Vegetation density impacts longwave enhancement by scaling the respective contributions of vegetation and atmosphere to sub-canopy longwave radiation as well as by governing the impact of meteorological forcing on vegetation temperatures (Todt et al., 2018). While observations have shown trunks heating up due to insolation and emission of longwave radiation consequently increasing (Rowlands et al., 2002; Pomeroy et al., 2009), diurnal variations in tree temperatures depend on exposure to insolation and thus vegetation density (Webster et al., 2016).

About a fifth of seasonally snow-covered land over the Northern Hemisphere is covered by boreal forests (Rutter et al., 2009), indicating the process of longwave enhancement affects a substantial fraction of global snow cover. Considerable challenges persist in the representation of snow cover and snowmelt in the current generation of climate models as historical simulations from Climate Model In-

tercomparison Project's fifth phase (CMIP5) underestimate observed trends and interannual variability of spring snow cover extent (SCE) (Derksen and Brown, 2012; Brutel-Vuilmet et al., 2013; Rupp et al., 2013; Mudryk et al., 2014; Thackeray et al., 2016). Snow Model Intercomparison Project's second phase (SnowMIP2) identified less skill in modelling snow for forested than for open sites, which was attributed to complex processes between atmosphere, snow, and vegetation (Essery et al., 2009; Rutter et al., 2009).

Among models displaying deficiencies in simulating snow cover evolution across boreal forests is the Community Land Model (CLM) version 4 and its parent Community Climate System Model version 4 (Thackeray et al., 2014, 2015). CLM uses a one-layer vegetation scheme and CLM version 4.5 (CLM4.5) has been found to show deficiencies in simulation of sub-canopy longwave radiation and longwave enhancement with overestimated diurnal cycles under clear-sky conditions (Todt et al., 2018). Similar issues have been mitigated in CLM4.5 by subdividing the roughness layer (Bonan et al., 2018) and in SNOWPACK, a one-dimensional snow cover model, by partitioning the canopy into two layers with separate energy balances and consequently separate vegetation temperatures (Gouttevin et al., 2015), which results in different longwave radiation fluxes emitted upward and downward from the vegetation.

In order to avoid implementing multiple canopy layers in a global land model and associated computational costs, this study presents an alternative guided by the effect of separate vegetation layers on sub-canopy longwave radiation. A correction to sub-canopy longwave radiation is implemented in CLM4.5 to reduce overestimated diurnal cycles, damping variations in longwave radiation emitted downward and, consequently, increasing variations in longwave radiation emitted upward. While simulation of sub-canopy longwave radiation and longwave enhancement by land surface models had so far been assessed using forest stand-scale forcing and evaluation data, this study uses land-only simulations of CLM4.5 and snow-off dates derived from global observations of snow water equivalent (SWE) to assess the impact of overestimated diurnal cycles in sub-canopy longwave radiation on simulated global snow cover and snowmelt. Therefore, this study has three objectives:

    i. To develop a correction of sub-canopy longwave radiation simulated by single-layer vegetation in CLM4.5;

    ii. To evaluate the effect of this correction on simulated diurnal cycles and daily averages of sub-canopy longwave radiation;

    iii. To quantify the impact of corrected sub-canopy longwave radiation on snow cover and snowmelt across the Northern Hemisphere.

Section 2 presents methodological details about treatment of sub-canopy longwave radiation in CLM4.5, the physical basis for the empirical correction, configuration of global land-only simulations, and calculation of snow-off date from SWE observations. Calculation of correction factors is detailed in Sect. 3, and their impacts on the simulated energy balance and seasonal cycle of snow cover are presented in Sect. 4. We conclude with a brief discussion in Sect. 5.

## 2 Methods

### 2.1 Sub-canopy longwave radiation in CLM4.5

Vegetation in CLM4.5 is parameterized as a single layer using a "big-leaf" approach (Oleson et al., 2013). Sub-canopy longwave radiation is calculated as the sum of atmospheric longwave radiation $LW_{atm}$ and longwave radiation emitted by vegetation $LW_{veg}$, weighted by vegetation emissivity $\varepsilon_v$:

$$LW_{sub} = (1 - \varepsilon_v)\, LW_{atm} + \varepsilon_v\, \sigma\, T_v^4 \tag{1}$$

using the Stefan-Boltzmann law with Stefan-Boltzmann constant $\sigma = 5.67\ 10^{-8}\ \mathrm{Wm^{-2}K^{-4}}$ and vegetation temperature $T_v$. Vegetation temperature is calculated based on an energy balance, net radiation minus turbulent heat fluxes. Radiative transfer of direct and diffuse shortwave radiation is calculated via a two-stream approximation (Sellers, 1985) considering one reflection from ground to canopy. Net longwave radiation is calculated from atmospheric longwave radiation, vegetation temperature, and (ground) surface temperature and determined by vegetation emissivity and emissivity of the ground. Calculation of turbulent heat fluxes in CLM4.5 is based on Monin-Obukhov similarity theory and described by Oleson et al. (2013). Vegetation emissivity depends on Leaf Area Index ($LAI$) and Stem Area Index ($SAI$) and is calculated as

$$\varepsilon_v = 1 - e^{-(LAI+SAI)}. \tag{2}$$

This parameter is a combination of emissivity in the physical sense and a weighing parameter based on vegetation density, however, we will stick to this denomination here for consistency with the nomenclature of the technical description of CLM4.5 (Oleson et al., 2013).

CLM4.5 subdivides grid cells based on land units, e.g. vegetated, glacier, or urban, and vegetated land units based on Plant Functional Types (PFTs), with up to 16 possible PFTs as well as bare soil. Sub-canopy longwave radiation is calculated for each PFT present in a grid cell, with separate values of LAI, SAI, and vegetation temperature for each PFT. All PFTs within one vegetated land unit share a single column of snow and soil, so that fluxes from vegetation to the ground are weighted averages over all PFTs. Consequently, changes in fluxes from an individual PFT affect snow cover beneath every PFT in a particular vegetated land unit.

## 2.2 Correction of sub-canopy longwave radiation in CLM4.5

For this study, a correction factor $f_{corr}$ was implemented in CLM4.5 to reduce unphysical diurnal variations in sub-canopy longwave radiation. As atmospheric longwave radiation is an input variable to CLM4.5, from either forcing datasets or the atmospheric component of CESM (Community Atmosphere Model, CAM), correction factors were used to scale longwave radiation emitted from vegetation:

$$LW_{sub} = (1 - \varepsilon_v)\,LW_{atm} + \varepsilon_v\,\sigma\,T_v^4\,f_{corr}. \qquad (3)$$

Conceptually, correction factors represent a vegetation structure consisting of multiple individual layers, so that longwave radiation fluxes emitted upward and downward from the vegetation are no longer equal by design. In a multi-layer canopy configuration, the uppermost layer contributes most to longwave radiation emitted upward to the atmosphere and directly absorbs incoming longwave and shortwave radiation fluxes. Conversely, the lowest layer contributes most to longwave radiation emitted downward to the surface, but is insulated from atmospheric fluxes by the canopy layers above.

Using this multi-layer canopy configuration as a guideline, longwave radiation emitted by the canopy was partitioned asymmetrically upward and downward in CLM4.5. The resulting above-canopy longwave radiation flux to the atmosphere was calculated as

$$
\begin{aligned}
LW_{above} = {} & \qquad\qquad\qquad\qquad\qquad\quad (4)\\
& (1 - \varepsilon_v)\,(1 - \varepsilon_g)\,(1 - \varepsilon_v)\,LW_{atm}\\
& + \varepsilon_v\left((2 - f_{corr}) + (1 - \varepsilon_v)\,(1 - \varepsilon_g)\,f_{corr}\right)\sigma\,T_v^4\\
& + (1 - \varepsilon_v)\,\varepsilon_g\,\sigma\,T_g^4
\end{aligned}
$$

with emissivity of the ground $\varepsilon_g$ and temperature of the ground $T_g$. Ground emissivity in CLM4.5 is calculated as a weighted sum of emissivities of snow (0.97) and soil (0.96), weighted by the fraction of snow covering a grid cell. In Eq. (4), the first term represents atmospheric longwave radiation transmitted through the vegetation, reflected by the ground, and transmitted through the vegetation to the atmosphere; the second term represents longwave radiation emitted from the vegetation reaching the atmosphere; and the third term represents longwave radiation emitted by the ground and transmitted through the vegetation to the atmosphere. The second term combines longwave radiation emitted by the vegetation directly to the atmosphere (first term in parentheses) and longwave radiation emitted downward from the vegetation, reflected by the ground, and transmitted through the vegetation to the atmosphere (second term in parentheses). For $f_{corr} > 1$, $LW_{above}$ decreases as reduction in the first term in parentheses $(2 - f_{corr})$ outweighs increase in the second term in parentheses $(1 - \varepsilon_v)\,(1 - \varepsilon_g)\,f_{corr}$, while $LW_{sub}$ in Eq. (3) increases. For $f_{corr} < 1$, $LW_{sub}$ decreases and $LW_{above}$ increases. Note that the sum of $LW_{sub}$

and $LW_{above}$ was not changed by the introduction of $f_{corr}$, which guaranteed conservation of energy. The calculation of vegetation temperature in CLM4.5 was not altered by this approach, so that the temperature of the single vegetation layer represented an average of multiple (theoretical) layers suggested by asymmetrical upward and downward longwave radiation fluxes.

## 2.3 Global offline simulations with CLM4.5

Offline simulations of CLM4.5 were forced by prescribed atmospheric data, using the CRUNCEP version 7 data set covering 1981 to 2016 and thus snow seasons 1981/82 to 2015/16 (Viovy, 2018). The impact of correction factors on longwave enhancement, snow cover, and snowmelt was assessed by comparing two simulations, a control run (henceforth, CTRL) and a run in which correction factors were implemented (henceforth, CORR). Correction factors were applied to evergreen needleleaf trees in CLM4.5 as given in Eq. (3) and Eq. (4). Two PFTs, Needleleaf Evergreen Boreal Trees (NEBTs) and Needleleaf Evergreen Temperate Trees (NETTs), represent evergreen forests across snow-covered areas in CLM4.5 and grid-cell coverage by these two PFTs is shown in Fig. 1a. Plant Area Index (PAI), the sum of LAI and SAI, is shown in Fig. 1b as a weighted average of NEBTs and NETTs.

## 2.4 Global observations of snow-off date

A blended data set of five global observation-based SWE products (henceforth, Blended-5) covering the period 1981 to 2010 (Mudryk et al., 2015) was used to estimate snow-off dates across the Northern Hemisphere and evaluate simulation of snowmelt in CTRL and CORR. In contrast to simulations, observations display snow persisting for physically unrealistic durations, which necessitates a SWE threshold to estimate snow-off dates (Krinner et al., 2018). While Mudryk et al. (2017) and Krinner et al. (2018) used thresholds of 4mm and 5mm, respectively, for estimates of spatial snow cover extent, a smaller SWE value was necessary to represent the precise timing of meltout within individual grid cells. A threshold of 1mm was used in this study to define meltout for the Blended-5 mean, and snow-off date was defined as the first day of a year for which SWE did not exceed this threshold. Sensitivity of snow-off dates to threshold values was tested for the range 0.5mm to 4mm, however, the overall conclusions of this study are unchanged for different thresholds.

## 3 Calculation of correction factors

Todt et al. (2018) created a "toy model", which utilized forest stand-scale forcing data to evaluate sub-canopy longwave radiation in CLM4.5 and revealed systematic simulation errors that depend on meteorological conditions. These me-

teorological conditions were categorized via insolation and cloudiness represented by effective emissivity of the sky, which is calculated as

$$\varepsilon_{sky} = \frac{LW_{atm}}{\sigma\, T_{air}^4} \tag{5}$$

using air temperature $T_{air}$. Based on those stand-scale simulations, this study calculated correction factor $f_{corr}$ from $\varepsilon_{sky}$ and insolation $SW_{in}$ as

$$f_{corr}^{-1} = b_0 + b_1\, \varepsilon_{sky} + b_2\, SW_{in} + b_3\, SW_{in}\, \varepsilon_{sky}. \tag{6}$$

Coefficients $b_{0,...,3}$ relate to the intercept of the equation, $\varepsilon_{sky}$, insolation, and interaction of $\varepsilon_{sky}$ and insolation, respectively, and were calculated via multiple linear regression from stand-scale simulation errors expressed as ratios (Fig. 2) and observations of $\varepsilon_{sky}$ and insolation at forest stands listed in Table 1. Consequently, correction factors were calculated as inverses of these ratios to scale longwave radiation in CLM4.5. For example, if stand-scale simulations revealed an overestimation of longwave radiation by 25% for particular values of $\varepsilon_{sky}$ and $SW_{in}$, correction factors in global simulations would be $1.25^{-1} = 0.8$ for the same meteorological conditions.

As CLM4.5 only simulates longwave radiation emitted from vegetation, simulation errors were calculated for $LW_{veg}$ that was derived from sub-canopy longwave radiation via Eq. (1) and Eq. (2) using measurements of atmospheric longwave radiation and PAI given in Table 1. Error ratios as a function of $\varepsilon_{sky}$ and insolation as well as estimates based on regression coefficients are shown in Fig. 2. Nighttime estimates are a linear function of $\varepsilon_{sky}$ as $SW_{in} = 0$, while daytime estimates include potential non-linear interactions of $\varepsilon_{sky}$ and $SW_{in}$. Both daytime and nighttime simulation errors generally increase in magnitude with clearer skies.

Regression coefficients as outlined in Eq. (6) are shown in Fig. 3 for every site and season, differentiated for day and night. Intercept $b_0$ and regression coefficient for $\varepsilon_{sky}$ $b_1$ agree in sign for all sites and agree in magnitude for all sites except Abisko (panels a, b, d, e), with little interannual variability for the two sites with multiple years of data (Alptal and Seehornwald). In contrast to Abisko, $b_0$ and $b_1$ for the deciduous forest at Cherskiy are similar to those for evergreen sites Alptal, Seehornwald, and Sodankylä despite featuring different vegetation type, structure, and density. Regression coefficients involving insolation agree in sign but differ in magnitude among Alptal, Cherskiy, Seehornwald, and Sodankylä (panels c and f), with similar values for the latter two sites due to little interannual variability for Seehornwald. In contrast, interannual variability is large for Alptal with higher magnitudes for all four years combined compared to Seehornwald and Sodankylä, while magnitudes are smallest for Cherskiy. For Abisko, five out of six regression coefficients display smallest magnitudes due to deciduous vegetation and

consequently low vegetation density as well as smaller simulation errors compared to other sites (Todt et al., 2018). Overall, uncertainties are largest for Abisko due to a short evaluation period, with no regression coefficient being significantly different from zero (or one as in the case of intercept $b_0$).

For implementation in global simulation CORR, regression coefficients were calculated based on one season each of Alptal, Cherskiy, Seehornwald, and Sodankylä in order to balance dense and sparser sites. Despite featuring a deciduous PFT, Cherskiy was included as regression coefficients are similar to evergreen sites. Individual seasons for Alptal, 2005, and Seehornwald, 2009, were chosen based on similarity of regression coefficients to those for all years combined of the respective site. Regression coefficients for these four sites combined are shown as red lines in Fig. 3. Estimates of simulation errors based on these regression coefficients are shown in Fig. 2 and explain 60% of variance in nighttime errors and 59% of variance in daytime errors.

## 4    Effect of correction in global simulations of CLM4.5

### 4.1    Sub-canopy longwave radiation – case study Alptal, Switzerland

For the location of Alptal, in contrast to other forest stands used in this study, forest stand and CLM4.5 grid cell feature similarly high vegetation densities (PAIs of 4.1 $m^2m^{-2}$ and 3.7 $m^2m^{-2}$, respectively) and thus similar vegetation emissivities $\varepsilon_v$ (0.983 and 0.975, respectively). This allows for a comparison of diurnal cycles of sub-canopy longwave radiation as well as longwave enhancement between stand-scale measurements and offline simulations. Implementation of correction factors in CLM4.5 results in decreased sub-canopy longwave radiation during daytime and increased sub-canopy longwave radiation during nighttime, thereby reducing diurnal cycles. For the grid cell representing Alptal, diurnal ranges decrease from about 70 $Wm^{-2}$ to about 30 $Wm^{-2}$ during snowmelt season (Fig. 4a and Fig. 4b). Observations at the forest stand show an average diurnal range of about 15 $Wm^{-2}$ during snowmelt season. Simulations and observations display a similar range of intraseasonal variability but do not agree in evolution and daily average of sub-canopy longwave radiation. Implementation of correction factors increases average sub-canopy longwave radiation, seen in Fig. 4b, for two reasons. Firstly, daytime correction depends on insolation, which changes throughout the snow cover season so that daytime correction varies to a higher degree than nighttime correction. Secondly, nights are longer than days prior to the boreal spring equinox, which leads to nighttime increases outweighing daytime decreases. Consequently, correction results in increased average sub-canopy longwave radiation even for equal magnitudes of daytime overestimation and nighttime underestimation.

Comparison of simulated and measured longwave enhancement is shown in Fig. 4c and Fig. 4d for Alptal. As for sub-canopy longwave radiation, the diurnal cycle of simulated longwave enhancement is reduced by implementation of correction factors with increased enhancement at night and decreased enhancement at daytime. Reduction of daytime longwave enhancement increases throughout the snowmelt season, which is due to increasing insolation and thus increasing reduction of sub-canopy longwave radiation during daytime. Longwave enhancement values vary between 1.1 and 1.4 in CTRL, which is predominately driven by diurnal cycles. The diurnal cycle of longwave enhancement is reduced by more than 50% in CORR, resulting in a diurnal range similar to observations and increased daily average longwave enhancement. Simulated longwave enhancement displays little intraseasonal variability, with variations mostly due to the overestimated diurnal cycle. This indicates that intraseasonal variability in sub-canopy longwave radiation is largely due to variations in atmospheric longwave radiation. In contrast, measured longwave enhancement values range from less than 1 to more than 1.6 and display little diurnal variability but high variability on synoptic timescales, which results in a different daily average of longwave enhancement compared to simulations. Moreover, lower average longwave enhancement for observations indicates more overcast conditions, which lead to smaller diurnal cycles in sub-canopy longwave radiation compared to simulations. Therefore, correction factors improve the realism of diurnal cycles of sub-canopy longwave radiation and longwave enhancement, encouraging usage for evaluation of impact on snow cover.

The contrast in variability between simulated and observed longwave enhancement can be seen in Fig. 5. Observations show a large range of longwave enhancement values that are closely tied to effective emissivity of the sky, which represents clear-sky (low $\varepsilon_{sky}$) and overcast (high $\varepsilon_{sky}$) conditions. Observed longwave enhancement increases for decreasing $\varepsilon_{sky}$ as the contrast between vegetation temperatures, increasing due to higher insolation, and effective temperature of the sky increases. Spread in observed longwave enhancement is small throughout the range of $\varepsilon_{sky}$, indicating little diurnal variability and the process of longwave enhancement depending on meteorological conditions. Simulations display a narrow range of $\varepsilon_{sky}$, which causes the lack of intraseasonal variability seen in Fig. 4c. The spread in simulated longwave enhancement values is substantially larger compared to observations for the respective range in $\varepsilon_{sky}$ representing overestimated diurnal cycles. Implementation of correction factors reduces the spread in longwave enhancement values and increases average longwave enhancement (see Fig. 4d), however, spread in longwave enhancement is still overestimated and average longwave enhancement is underestimated in CORR compared to observations for the respective range in $\varepsilon_{sky}$.

## 4.2 Longwave enhancement and limited spatial variability in $\varepsilon_{sky}$

Lack of variability in simulated $\varepsilon_{sky}$, as seen for the grid cell of Alptal, across the Northern Hemisphere results in spatially similar correction factors that largely dependent on insolation. However, variability in both insolation and diurnal ranges of atmospheric longwave radiation indicate small variations in meteorological forcing that are not represented by $\varepsilon_{sky}$. Therefore, $\varepsilon_{sky}$ in simulations may indicate clear-sky conditions even when insolation and atmospheric longwave radiation suggest more overcast conditions, resulting in overestimated correction factors and overcorrection of sub-canopy longwave radiation. This overcorrection results in larger nighttime than daytime values of sub-canopy longwave radiation in contrast to atmospheric longwave radiation and occurs mostly along continental coasts (Fig. 6). Consequently, a contour line is used in the following to denote an overcorrection for 10% of days.

To demonstrate the impact of correction factors spatially, maps of longwave enhancement beneath evergreen needleleaf forests in CLM4.5 are shown in Fig. 7a and Fig. 7b. Averages over boreal winter and spring show an enhancement of longwave radiation beneath canopies by about 20% to 30% and display little differences across boreal forests, which is due to small spatial variability in both $\varepsilon_{sky}$ and vegetation density (Fig. 1). CORR displays increased average longwave enhancement north of 40°N with an additional enhancement of longwave radiation of up to 5% beneath dense boreal forests. Changes in longwave enhancement generally increase with latitude as daytime correction factors vary with insolation while nighttime correction factors are independent of latitude. A higher increase in longwave enhancement can be seen for higher vegetation density within regions covered by boreal forests (Fig. 1b) due to weighting of contributions to subcanopy longwave radiation (Eq. (3)).

## 4.3 Snow cover and snowmelt

Changes in sub-canopy longwave radiation induced by the correction increase the net energy flux to the surface, which can be seen for grid cell-averaged snow surface temperature (Fig. 7c and Fig. 7d). Simulated average snow surface temperatures are determined by latitude, topography, and continentality, reaching values of less than -40°C in the mountainous regions of northeastern Siberia (Fig. 1c), and range between -20°C and -15°C for boreal forests, the outlines of which can be seen in central Siberia and central North America. The impact of correction factors is limited to grid cells for which vegetation is dominated by evergreen needleleaf trees and implementation results in an increase in average snow surface temperature of up to 2°C. The lack of spatial variability is caused by little spatial variability in meteorological conditions, high vegetation density, and similarly high PFT coverage across boreal forests (Fig. 1a and Fig. 1b).

Cold content, the energy required to raise snow temperatures to $0°C$, is used to quantify the impact of correction factors on the entire snow column. Average cold content simulated by CLM4.5 mostly reaches values of up to $4\,\mathrm{MJm^{-2}}$ and exceeds $5\,\mathrm{MJm^{-2}}$ only in glaciated grid cells (Fig. 7e). In CTRL, simulated average cold content ranges between $1.5\,\mathrm{MJm^{-2}}$ and $3\,\mathrm{MJm^{-2}}$ across boreal forests, with lowest values in northeastern Europe and highest values in eastern Siberia, western Canada, and Quebec. Relative changes in cold content from CTRL to CORR display spatial differences with cold content generally decreasing across boreal forests (Fig. 7f). Reductions in average cold content reach up to 30% in northeastern Europe and western North America and up to 20% in central North America. Across Siberian boreal forests, relative reductions decrease from west to east from more than 20% to about 10%. Spatial differences in relative reductions correspond to spatial differences in average cold content, with higher relative reductions for smaller averages, representing a more even spatial pattern of absolute reductions in cold content as indicated by changes in snow surface temperature (Fig. 7d).

Spatial patterns in snow-off date are similar to those in cold content with higher cold content corresponding to later meltout (Fig. 7g and Fig. 7h). Changes in snow-off date from CTRL to CORR display stark spatial contrasts with meltout happening up to 10 days earlier in central Europe and on the western coast of North America. Meltout is advanced by about 5 days for boreal forests in northeastern Europe and western Siberia and slightly less for boreal forests in central North America. In contrast, meltout is delayed in mountains of southeastern Siberia (Fig. 1c), where meltout occurs late among boreal forests.

As offline simulations lack spatial variability in $\varepsilon_{sky}$, latitude (through insolation) and duration of snow on the ground (through day length) control spatial differences in impact of correction of sub-canopy longwave radiation on snow-off date. Changes in longwave enhancement due to correction of sub-canopy longwave radiation before and after the boreal spring equinox, approximated by averages over February/March and April/May, display opposite signs across the Northern Hemisphere (Fig. 8), with shorter (longer) days than nights before (after) the equinox resulting in an increase (decrease) in daily average longwave enhancement. Generally, lower insolation at higher latitudes leads to a more positive impact of correction on daily average longwave enhancement, increasing (decreasing) positive (negative) changes in longwave enhancement with increasing latitude before (after) the boreal spring equinox. Across mid-latitudes, increase in daily average longwave enhancement over February and March is roughly similar to decrease in daily average longwave enhancement over April and May, while increase over February and March outweighs decrease over April and May across high latitudes including most of the regions covered by boreal forests.

Reasons for spatial differences in changes of snow-off date across Siberian boreal forests are explored in Fig. 9. Snow-off dates are similar spatially in CTRL, likely caused by higher elevations in southeastern Siberia compensating for less cold content, and meltout generally occurs past the boreal spring equinox in northwestern and southeastern Siberia. However, higher insolation for southeastern Siberia results in higher reductions of daytime sub-canopy longwave radiation by correction factors and consequently smaller increases in daily average sub-canopy longwave radiation prior to the boreal spring equinox compared to northwestern Siberia. Although changes in sub-canopy longwave radiation are still positive in southeastern Siberia accumulated over the snow season, causing a decrease in cold content, reduction in daily average sub-canopy longwave radiation by correction factors past the boreal spring equinox cancels out the previous increase and consequently, snowmelt is slightly delayed. In contrast to southeastern Siberia, meltout is slightly accelerated in central North America although both latitude and meltout date are similar, as relative reductions in cold content are generally higher. However, differences in changes in meltout date between central North America and southeastern Siberia are minor.

## 4.4 Snow-off date in comparison to observations

Simulated and observed snow-off dates are compared in Fig. 10 for grid cells with consistent snow cover throughout preceding December and coverage by evergreen needleleaf trees of at least 50%. Simulations CTRL and CORR generally feature a narrower probability density function (PDF) of snow-off dates, indicating a shorter snowmelt season, and later meltout compared to observations across the entire Northern Hemisphere (Fig. 10a). While shapes of observed PDFs are well represented by simulations over Eurasia (Fig. 10b, d), observations show a clearer, shorter peak of meltout compared to simulations over mountainous western North America (Fig. 10c). Correction of sub-canopy longwave radiation displays little impact when accumulated over the entire Northern Hemisphere, however, it systematically reduces the delay of simulated snow-off dates throughout the snowmelt season. PDFs of snow-off dates for regional subsets reflect spatial patterns seen in Fig. 7h, with minor differences between CTRL and CORR over most of western North America (Fig. 10c) and eastern Siberia (Fig. 10d) but substantial acceleration of snow-off dates over western Siberia and eastern Europe (Fig. 10b) due to correction of sub-canopy longwave radiation.

The regionally limited impact of corrected sub-canopy longwave radiation is highlighted by filtering PDFs of snow-off date for grid cells with average differences in snow-off date between CORR and CTRL of at least 3 days (Fig. 10e, f). Correction of sub-canopy longwave radiation improves timing of meltout in filtered grid cells, especially over western Siberia and eastern Europe where the filtered PDF for

CORR, in contrast to CTRL, closely resembles observations. PDFs of snow-off dates derived from Blended-5 SWE display sensitivity to threshold choices, however, this uncertainty is generally smaller than differences between simulations and observations.

## 5   Discussion

Todt et al. (2018) found roughly similar magnitudes for daytime overestimations and nighttime underestimations of sub-canopy longwave radiation in CLM4.5; however, this study shows that different durations of day and night over the snow cover season result in a net positive impact of correction on daily averages of sub-canopy longwave radiation. Correction factors change throughout the snowmelt season due to increasing insolation and length of day. Consequently, net impact on daily averages of sub-canopy longwave radiation varies resulting in spatial differences in impact on cold content over the snow cover season and meltout date. Net increase in sub-canopy longwave radiation during the snow cover season is highest for regions of early snowmelt where snow is already comparatively warm, which results in accelerated snowmelt. Lundquist et al. (2013) showed that forests enhance snowmelt compared to open area in regions where winters are warm and mid-winter melt events happen, during which longwave enhancement outweighs shading. Spatial differences in change of meltout date broadly agree with this pattern as the highest acceleration of melt occurs for regions with warmer winters as indicated by snow surface temperatures (Fig. 7c), suggesting that mid-winter melt events could be underestimated by CLM4.5. Conversely, correction of sub-canopy longwave radiation results in slightly delayed snowmelt in southeastern Siberia albeit average cold content over the entire snow cover season being reduced. This delay is due to meltout happening substantially later than the boreal spring equinox and high insolation during the snowmelt period, which result in reduction in daytime sub-canopy longwave radiation outweighing increased sub-canopy longwave radiation during night. Consequently, overestimated diurnal cycles of sub-canopy longwave radiation in CLM4.5 lead to spatial differences in impact on snowmelt timing across boreal forests in offline simulations.

Previous comparison between offline simulations of CLM4 and observations have shown CLM4 failing to accurately simulate the timing of both snow ablation and snow accumulation across boreal forests, with snowmelt compressed into the period March to May (Thackeray et al., 2014, 2015). This shortened snowmelt season is confirmed by comparison of offline simulations of CLM4.5 with global observations, and correction of sub-canopy longwave radiation is found to have only minor impact on this deficiency. However, offline simulations also display a delay in snow-off dates compared to observations, which is decreased by correction of sub-canopy longwave radiation. This impact is small when considered over the entire Northern Hemisphere, but its importance varies regionally. For example, correction of sub-canopy longwave radiation substantially improves simulated snowmelt timing over western Siberia, which suggests overestimated diurnal cycles in sub-canopy longwave radiation are a contributing factor to delayed snowmelt in offline simulations of CLM4.5. Thackeray et al. (2014, 2015) also showed SCF increasing earlier than observed across boreal forests in CLM4 and, although this study does not focus on the snow accumulation period, processes governing the influence of correction factors are the same as for the snow ablation period. As most snowfall occurs past the boreal autumn equinox, when daily average sub-canopy longwave radiation is increased due to correction factors, correction could delay the accumulation of snow across boreal forests. Therefore, overestimated diurnal cycles in sub-canopy longwave radiation also potentially contribute to this deficiency in the simulation of snow cover timing.

Changing seasonality in a warming climate may have implications for snowmelt and longwave enhancement. Future warming will lead to earlier snowmelt, when less energy from insolation is available for melt, which will likely result in lower melt rates (Musselman et al., 2017). A shortened snow season indicates more asymmetrical lengths of day and night during snowmelt and consequently, overestimated diurnal cycles of sub-canopy longwave radiation in CLM4.5 could result in even higher underestimations in daily averages. Moreover, underestimated sub-canopy longwave radiation suggests that CLM4.5 underestimates melt rates in general. In turn, future projections are complex, as corrected and thus increased sub-canopy longwave radiation might cancel out reduced energy from insolation due to earlier snowmelt. Nonetheless, the contribution of longwave enhancement to snowmelt is likely to increase in the future, further necessitating accurate simulation of sub-canopy longwave radiation.

Implementation of correction factors resulted in realistic average diurnal ranges of sub-canopy longwave radiation and longwave enhancement, but more substantial underestimation than overestimation of longwave enhancement seen in Fig. 5 suggests that the impact of shortcomings in CLM4.5 on snow cover and snowmelt might still be underestimated by this study. Gouttevin et al. (2015) and Todt et al. (2018) have shown the implementation of biomass heat storage to result in a net positive impact on sub-canopy longwave radiation as well as a slight reduction of diurnal cycles. This suggests that heat storage by biomass could further reduce nighttime underestimation in CLM4.5 and improve the simulation of sub-canopy longwave radiation and longwave enhancement.

## 6   Conclusions

This study assessed the impact of deficiencies in simulated longwave enhancement by forest canopies on snow cover in CLM4.5. Sub-canopy longwave radiation simulated by

CLM4.5's single-layer vegetation was corrected based on the damping effect of multiple canopy layers. Correction factors were derived from forest stand-scale simulations and subsequently implemented for evergreen needleleaf trees in global land-only simulations of CLM4.5. Correction reduces overestimated diurnal cycles of sub-canopy longwave radiation by decreasing daytime overestimations and nighttime underestimations. This results in a net increase of sub-canopy longwave radiation over the entire snow cover season, due to longer nights than days. Consequently, correction results in increasing average snow temperatures and earlier meltout, indicating that CLM4.5 underestimates snow temperatures and delays snowmelt due to overestimated diurnal cycles of sub-canopy longwave radiation. Comparison with observations confirmed a delay of meltout in land-only simulations of CLM4.5 across boreal forests, which is decreased by correction of sub-canopy longwave radiation. While land-only simulations exhibit a spatially uniform underestimation of snow temperatures by CLM4.5 across evergreen boreal forests, the impact of correction on meltout timing displays spatial differences that depend on insolation and duration of snow on the ground. The effect of overestimated diurnal cycles on daily average sub-canopy longwave radiation changes throughout the snowmelt season as insolation and length of day increase. Consequently, CLM4.5 delays snowmelt more in regions of warmer snow cover and earlier meltout. However, spatial variability in impact on snow cover is limited in land-only simulations of CLM4.5 due to a lack of variability in meteorological conditions.

*Code and data availability.* Code is available on GitHub at https://github.com/mtodt/2018_OfflineSimulations in order to derive correction factors, implement correction factors in CLM4.5, post-process simulations, and create figures shown in this study. Meteorological observations for forest stands are available as follows: (1) on GitHub at https://github.com/mtodt/2018_ToyModel for Alptal and Seehornwald; (2) from the British Atmospheric Data Centre at http://catalogue.ceda.ac.uk/uuid/9c8c86ed78ae4836a336d45cbb6a757c for Sodankylä and http://catalogue.ceda.ac.uk/uuid/6947880b98d32e249a8638ebe768efd2 for Abisko; and (3) from the Arctic Data Center at https://arcticdata.io/catalog/view/doi:10.18739/A2BG2H890 for Cherskiy. Forest stand-scale simulations were performed by Todt et al. (2018) and code is available on GitHub at https://github.com/mtodt/2018_ToyModel. The Blended-5 product of daily observed snow water equivalent is available from the National Snow and Ice Data Center at http://nsidc.org/data/nsidc-0668.

*Author contributions.* MT, NR, and CF designed the experiments. MT and LW performed the simulations and MT analyzed them. MT prepared the manuscript with contributions from all co-authors.

*Competing interests.* The authors declare that they have no conflict of interest.

*Acknowledgements.* This research was supported by the Canadian Sea Ice and Snow Evolution (CanSISE) Network, which is funded by the Natural Science and Engineering Research Council of Canada's Climate Change and Atmospheric Research program. Funding was also provided by the US National Science Foundation grant PLR-1417745 and The Picker Interdisciplinary Science Institute at Colgate University through Mike Loranty. We want to thank Tobias Jonas and Clare Webster for providing data from Alptal and Seehornwald as well as Heather Kropp and Mike Loranty for their support using data from Cherskiy. We would also like to thank two anonymous reviewers whose helpful comments improved this paper.

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

**Table 1.** Forest stands used for calculation of correction factors based on simulations by Todt et al. (2018). Vegetation density is given as Plant Area Index (PAI), the one-sided area of plant components per unit ground surface area including stems, branches, and leaves or needles. Abisko and Cherskiy feature deciduous vegetation, so that trees were leafless throughout the simulation periods and PAI values do not consider leaves or needles.

| Location | Abisko, Sweden | Alptal, Switzerland | Cherskiy, Russia | Seehornwald, Switz. | Sodankylä, Finland |
|---|---|---|---|---|---|
| **Latitude [°N]** | 68.4 | 47.1 | 68.7 | 46.8 | 67.4 |
| **Longitude [°E]** | 18.8 | 8.8 | 161.4 | 9.9 | 26.6 |
| **Snowmelt Season** | 2011 | 2004-07 | 2017 | 2008-12 | 2012 |
| **Days of Evaluation** | 9 | 41, 57, 73, 85 | 51 | 116, 90, 106, 83, 116 | 37 |
| **Tree Type** | birch | fir & spruce | larch | fir & spruce | pine |
| **Tree Height [m]** | 3.5 | 25 | 5 | 25 | 18 |
| **PAI [$m^2 m^{-2}$]** | 0.44 | 4.1 | 0.67 | 5.1 | 1.14 |

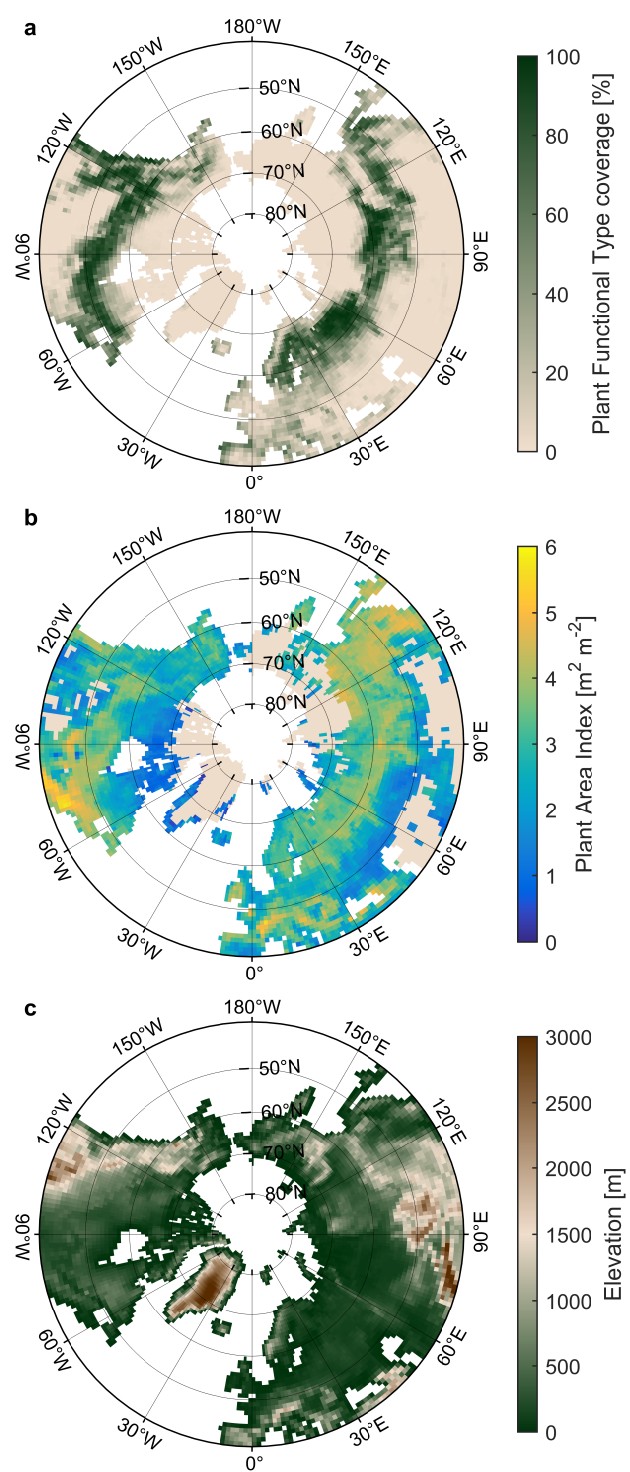

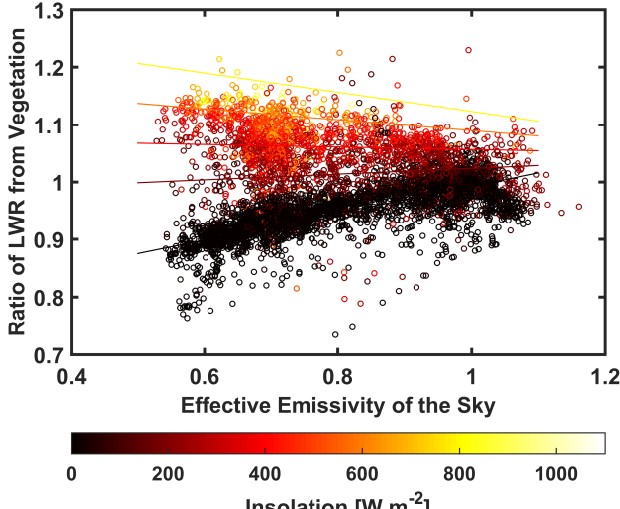

**Figure 2.** Ratio of longwave radiation emitted from vegetation simulated by CLM4.5 and estimated from forest stand observations as a function of effective emissivity of the sky (abscissa) and insolation (colour) for Alptal (season 2005), Seehornwald (season 2009), Sodankylä, and Cherskiy. Lines represent solutions of Eq. (6) for multiple values of insolation: 0, 200, 400, 600, and 800 $\mathrm{W\,m^{-2}}$.

**Figure 1.** Coverage of vegetated landunit within grid cell by combination of Needleleaf Evergreen Boreal Trees (NEBTs) and Needleleaf Evergreen Temperate Trees (NETTs) (a), Plant Area Index (PAI) for combination of NEBTS and NETTs weighted by PFT fractions (b), and grid-cell average elevation (c) based on CLM4.5's $0.9° \times 1.25°$ surface dataset.

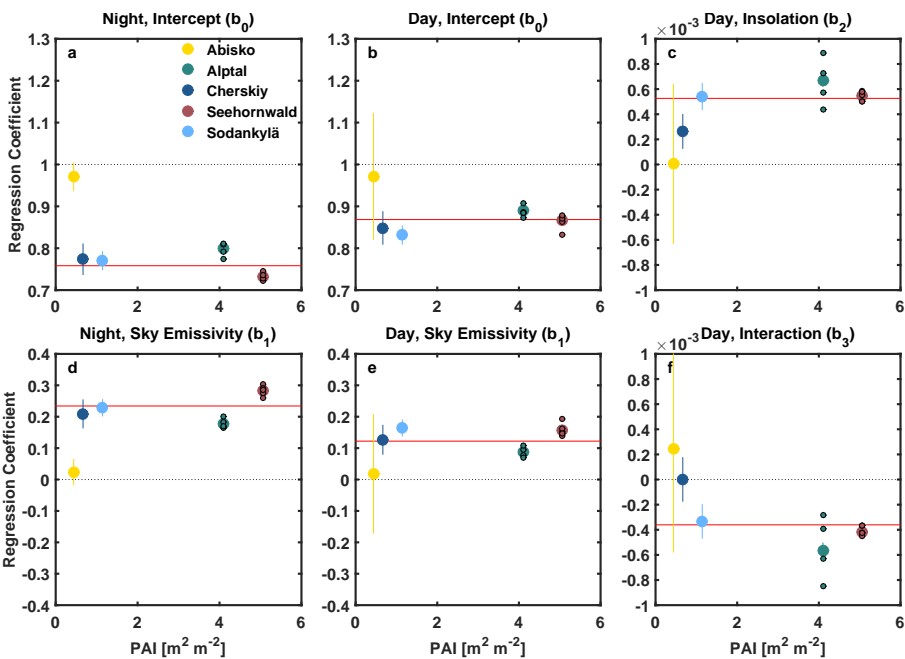

**Figure 3.** Regression coefficients (Eq. (6)) for forest stands at Abisko (yellow), Alptal (green), Cherskiy (dark blue), Seehornwald (maroon), and Sodankylä (light blue) with small circles indicating individual seasons for Alptal and Seehornwald and solid lines indicating 95%-confidence intervals. Red lines display regression coefficients calculated from a combination of Alptal season 2005, Cherskiy, Seehornwald season 2009, and Sodankylä. Intercept $b_0$ and regression coefficient for $\varepsilon_{sky}$ $b_1$ are differentiated for night (a and d, respectively) and day (b and e, respectively). Regression coefficient for insolation $b_2$ and regression coefficient for interaction of $\varepsilon_{sky}$ and insolation $b_3$ are shown for day only (c and f, respectively). Regression coefficients involving insolation have the unit $\mathrm{W^{-1}\,m^2}$.

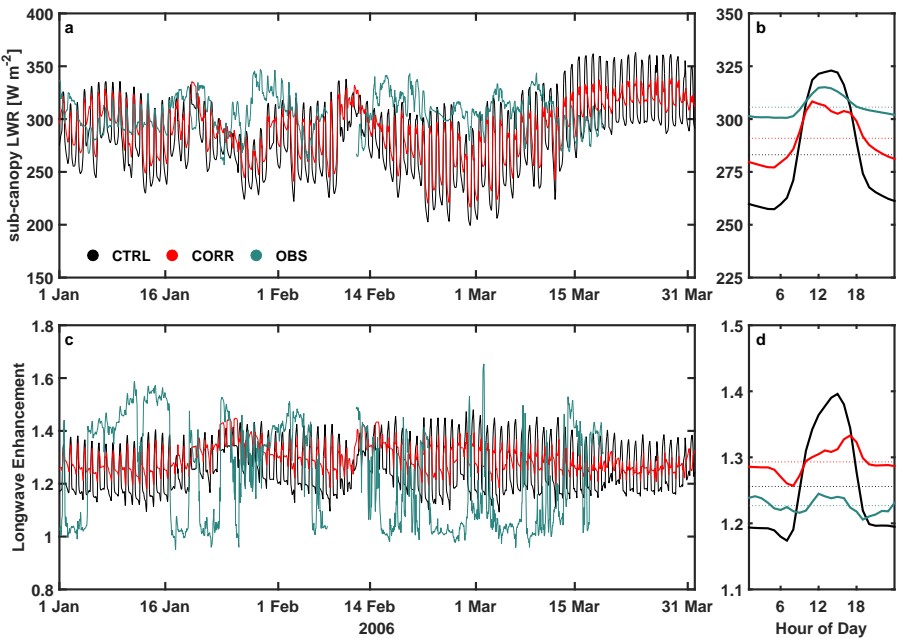

**Figure 4.** Hourly time series (a, c), diurnal cycles (solid in b, d), and JFM averages (dotted in b, d) of sub-canopy longwave radiation (a, b) and longwave enhancement (c, d) for the snowmelt season in 2006 at Alptal, Switzerland. Measurements at the forest stand (green) are shown for comparison with offline simulations CTRL (black) and CORR (red) for boreal evergreen needleleaf trees in the corresponding gridcell of Alptal, Switzerland. Gaps in measurements are due to quality checks and excluded from calculation of diurnal cycle and JFM average.

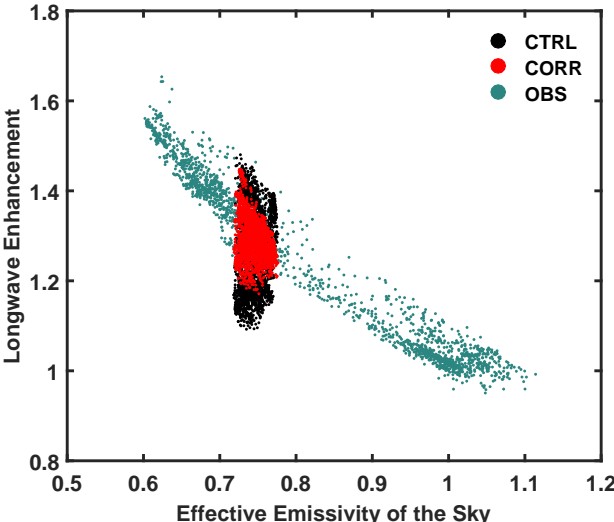

**Figure 5.** Longwave enhancement measured (green) at the forest stand of Alptal, Switzerland and simulated in CTRL (black) and CORR (red) for boreal evergreen needleleaf trees in the corresponding gridcell of Alptal, Switzerland as a function of effective emissivity of the sky. Each data point represents an hourly average seen in Fig. 4c.

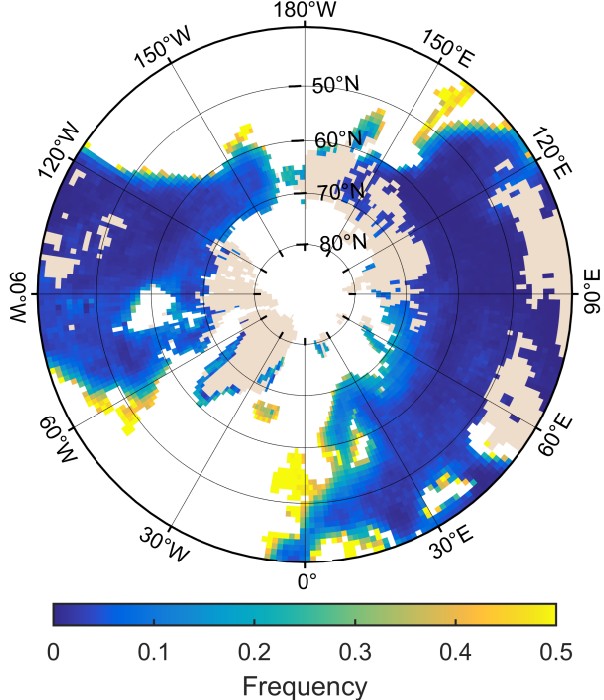

**Figure 6.** Frequency of days for 2004 - 2007 during which implementation of correction factors results in higher nighttime than daytime sub-canopy longwave radiation despite higher daytime than nighttime atmospheric longwave radiation.

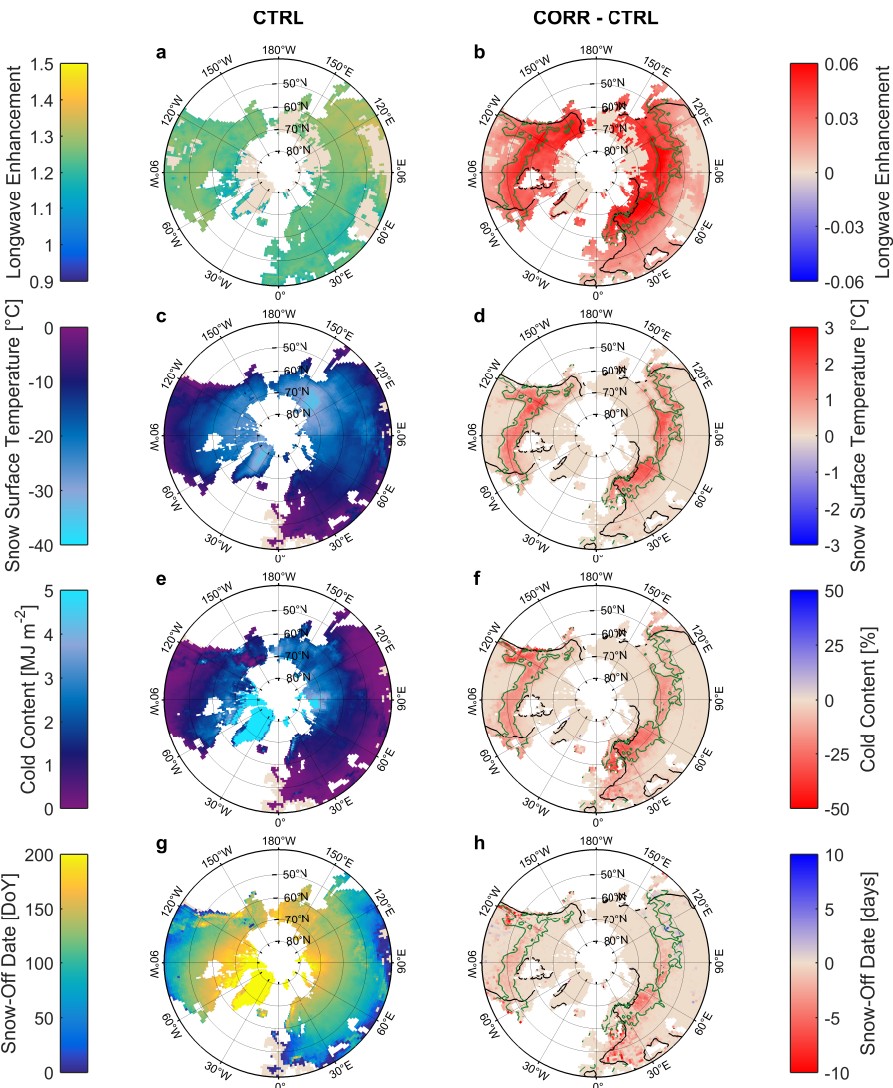

**Figure 7.** Averages in CTRL (a, c, e, g) and differences between CORR and CTRL (b, d, f, h) for longwave enhancement beneath evergreen needleleaf trees (a, b), snow surface temperature (c, d), cold content (e, f), and snow-off date (g, h). Longwave enhancement is averaged over December to May while snow surface temperature and cold content are averaged over entire snow cover seasons. Differences CORR - CTRL are calculated as averages of differences between each individual snow cover season. For panels c-h, a mask is applied to filter out grid cells that are not perennially snow-covered. Black lines demarcate continental areas with less than 10% of overcorrected days. Green lines demarcate areas with coverage by evergreen needleleaf trees of at least 50%.

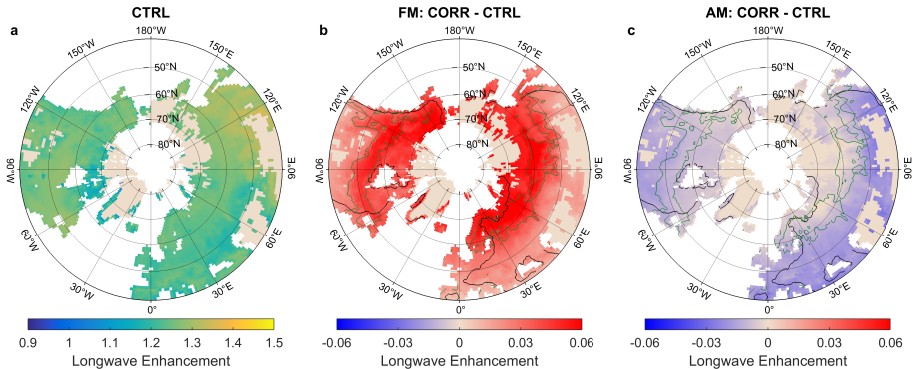

**Figure 8.** Longwave enhancement beneath evergreen needleleaf trees as average over December to May in CTRL (a, as in Fig. 7) and as difference between CORR and CTRL over February and March (b) and over April and May (c) . Differences CORR - CTRL are calculated as averages of differences between each individual year. Black lines demarcate continental areas with less than 10% of overcorrected days. Green lines demarcate areas with coverage by evergreen needleleaf trees of at least 50%.

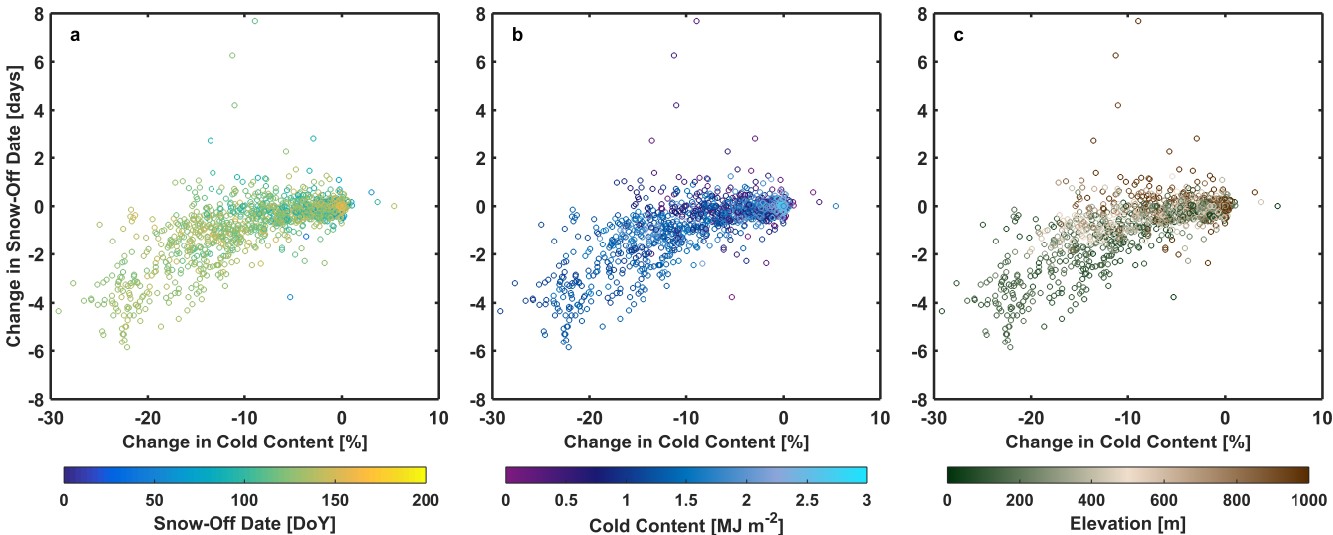

**Figure 9.** Change in cold content and snow-off date from CTRL to CORR as a function of (a) snow-off date and (b) cold content in CTRL as well as elevation (c) for grid cells within the area 40°E to 140°E and 42°N to 70°N.

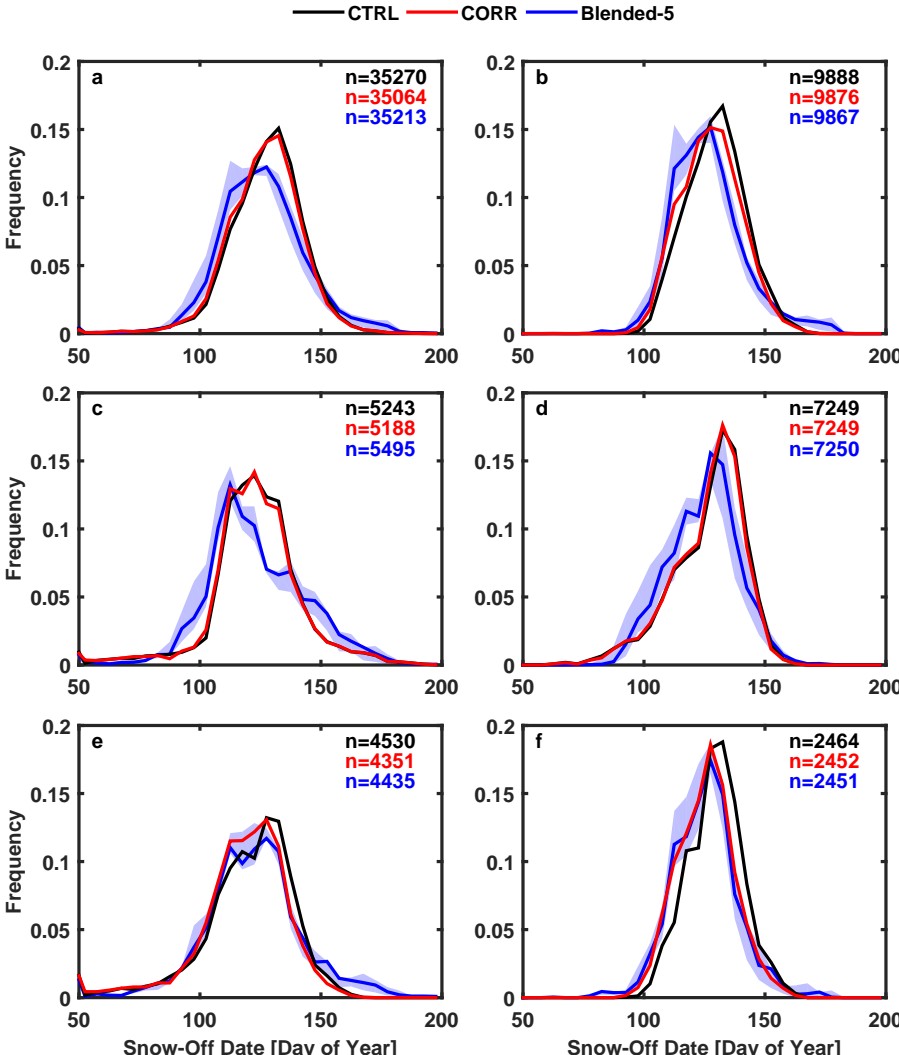

**Figure 10.** PDFs of snow-off dates and sample sizes *n* for CTRL (black), CORR (red), and observations (blue) over 1982-2010 across grid cells with coverage by evergreen needleleaf trees of at least 50% and snow cover persisting throughout December. Observational estimates are shown for SWE thresholds of 1mm (bold line) and 0.5mm to 4mm (shaded area). Panels show entire Northern Hemisphere (a), eastern Europe and western Siberia (b, 29.5°E to 90.5°E and 49°N to 66°N), western North America (c,104.5°W to 125.5°W and 39.5°N to 66°N), and eastern Siberia (d, 90.5°E to 135.5°E and 44°N to 66°N). Panels e and f are as panels a and b, respectively, but only for grid cells with average changes in snow-off dates of at least 3 days (as seen in Fig. 7h).