# Peer review of "Simulated single-layer forest canopies delay Northern Hemisphere snowmelt"

_The Cryosphere, 2018_

## Referee Comment (RC1) · Anonymous Referee #1 · 21 Jan 2019

Review of Simulated single-layer forest canopies delay Northern Hemisphere snowmelt by Markus Todt, Nick Rutter, Christopher G. Fletcher and Leanne M. Wake

This study assessed the effects of simple correction factors for partioning above and below simulated longwave radiation of forest canopies. These corrections were implemented for evergreen needleleaf trees in the off-line land-only simulations of the Community Land Model CLM4.5. Its impacts were tested on snow cover melting change.

Correction derived from forest stand-scale simulations (5 sites) results in a net increase of sub-canopy longwave radiation over the entire snow cover season leading in increasing average snow temperatures. This expands snowmelt duration across boreal forests by accelerating early snowmelt and delaying late snowmelt, in agreement with some previous studies.

However, the validation of the proposed improvements is not clearly demonstrated, using for example independent external database such as Globsnow2, including daily snow cover extent evolution, and/or MODIS snow cover products.

For the near future for validation and/or assimilation, the authors could mentioned the new radar Sentinel datasets that allow to monitor the wet snow evolution through open forest canopy (1). For closed forest area, the uncertainties seem still important?

Specific comments Please clarify how you estimate the "Longwave enhancement" parameter? Why do you differentiated day to night sky emissivities? (Fig. 3 d and e) Which ground (or snow?) emissivity value do you consider?

(1) Small, David; Rohner, Christoph; Miranda, Nuno; Rüetschi, Marius; Waser, Lars; Vögtli, Marius; Schaepman, Michael E(2018). Level 3 wide-area backscatter time-series for wet-snow mapping and forest classification. In: EGU General Assembly 2018, Vienna, 8 April 2018 - 13 April 2018.

––––––––––––––––––––––––––––––

---

## Referee Comment (RC2) · Anonymous Referee #2 · 28 Feb 2019

The authors examine the large-scale effets of a bias affecting the subcanopy LW radiation simulation in the CLM4.5 coupled model. One of these effects is the lengthening of the snow cover duration in the most of the Northern Hemisphere boreal forested regions, which is related to a severe night-time negative bias in subcanopy LW that dominates during the winter period prior to equinox. The authors show that partial correction of this bias, that intends to reproduce the behavior of a two-layer canopy model, enhances the energy delivered to the subcanopy snowpack and therefore also enhances snowmelt.

The paper is rather well written, and tackles an issue of clear significance for global climate modelling.
However, in my opinion, it suffers from two weakesses, that currently undermines its potential impact in the climate community :
1.
Technically, the demonstration is not thoroughly made, that single-layer canopy formulations generate melt delay in the NH with respect to « real world » (that here would be observations). Indeed, the demonstration of this effect is only made with respect to an approximation of a 2-layer model, that may itself be heavily biased. This is all the more worrying as Todt et al., 2018, illustrate an increase in a subcanopy LW positive bias upon the use of a 2-layer model at the Seehornwald conifer site (Todt et al., 2018, Figure 5). The melt delay associated with 1-layer canopy models claimed by the authors, may clearly be real, but the demonstration should be improved, for instance by confrontation of simulation results to (i) in-situ data at field sites and (ii) satellite estimate of NH snow disappearance dates. Comparison of simulation results to observed snow-cover fractions is briefly mentioned in the Discussion while it should be an important part of the Result section (as it is already quite well advertised as a rationale for this study in the Introduction). Thackeray et al., 2015 (their Figure 3 for instance) provides a good baseline for such comparisons.
2.
Secondly, for the evaluation of the delay effect against in-situ data, simulation errors coming from the meteorological forcing should be minimized. Evergreen forest sites used in Todt et al., 2018, provide appropriate observed meteorological data. They could be used in the place of erroneous large-scale meteorological forcing data (for e.g. Figure 4 and the associated results analysis).

Additionally, please find below some other, more minor comments on this work :

p1L10 : the last sentence of the abstract associates « boreal forests » and « warm winters » where « snowmelt occurs early ». This is not very intuitive
p2L3 : cite Krinner et al., 2018
p3L9-12 : « emissivity » is unappropriately used. What the authors call « emissivity », is an unexplained combination of emissivity and sky view factor. Please explicit and justify the approximations that you make here.
p3L9-12 : please explain briefly how Tv is calculated.
P6L11-13 : In the current structure of the paper, calculating and exhibiting correction factors for deciduous forest regions makes in my opinion little sense, as no use is made of them, and very little analysis of their difference w/r to coefficients calculated for evergreen forests is made. If Cherskii's coefficients are similar to those from evergreen sites, then what is the added value of including this site in the calibration intended for evergreen sites ?! To me, it just undermines the calibration approach.
Calculating correction coefficients for deciduous forests stands may be interesting, though, providing their difference (w/r evergreen sites) and impact (on e.g. snowmelt) is discussed in the paper.
p6L20 : please add 'CLM4.5' before 'grid cell' for more clarity
P6paragraph4.1 : I suggest to carry this analysis using an observed forcing to evidence the effect of going from 1-layer to 2-layer w/r to observations in an « ideal » case. Also, it should be mentionned

more clearly in the text that the selected period corresponds to the snowmelt season at Alptal and hence is relevant to assess the effect of subcanopy LW on snowmelt.

P7Paragraph4.2 : Maybe other forcings than CRUNCEP, exhibit less bias with respect to spatio-temporal in sky emissivity. Brun et al., 2013, concluded that ERA-i generally leads to improved simulations (w/r to other forcings) over large areas of the N high latitudes.

P8L3 : I suggest to outline the regions wth frac_PFT>0.5 on these maps, for better understanding of the effects of CORR and their magnitude.

P8L22:maybe the glaciated areas should be masked out as they are not the focus of this study. Otherwise, the question arises as to whether cold content for these areas refers to the snow, or to the whole snow+ice columns.

P8L30 : this is not true to my understanding as melt out date is largely determined by the energy required to melt the snow (which is often higher than the one needed to raise snow temperature to 0°C)

P9L1 : maybe the delay between melt-out-date, and equinox, could be an interesting additionnal explanatory variable in Fig8. Also, an illustation of daily cycle changes before and after equinox for Southeastern regions would be a great complement to the explanations.

P9L15 : specify « over the study region »

P9L30 : lllustrations of comparisons to observations like in Thackeray et al. 2014, 2015, is exactly what is missing in your study, and would add great value to it.

Overall, there is a lack of proper illustration of the competing effects of CORR on the daily subcanopy LW radiation before vs after the equinox, and how this governs the effects of CORR at the global scale.

References

Brun, E., Vionnet, V., Boone, A., Decharme, B., Peings, Y., Valette, R., ... & Morin, S. (2013). Simulation of northern eurasian local snow depth, mass, and density using a detailed snowpack model and meteorological reanalyses. Journal of Hydrometeorology, 14(1), 203-219.

Krinner, G., Derksen, C., Essery, R., Flanner, M., Hagemann, S., Clark, M., ... & Ménard, C. B. (2018). ESM-SnowMIP: assessing snow models and quantifying snow-related climate feedbacks. Geoscientific Model Development, 11, 5027-5049

---

## Editor Comment (EC1) · Florent Dominé (Editor) · 1 Mar 2019

Dear Authors,

The topic of your paper is very interesting, but both reviewers have significant reservations regarding the strength of your approach. In particular, it is essential to demonstrate the reality of the alleged bias using field or remote sensing data. It is critical that you convincingly address this comment before your paper can be considered for publication in The Cryosphere. I do encourage you to submit a revised version, but it will probably be sent again for review.

Thank you for submitting your work to The Cryosphere.

Best regards

[Figure]

Florent Domine, Editor

---

## Author Comment (AC1) · 20 Jun 2019

This document contains comments by reviewers (regular font), **responses to reviewers (bold)**, and *citations from the manuscript (italic)* including additions to the manuscript (blue).

Reviewer 1 Comments:

...However, the validation of the proposed improvements is not clearly demonstrated,

using for example independent external database such as Globsnow2, including daily snow cover extent evolution, and/or MODIS snow cover products.

**It is true that a validation of the impact of corrected sub-canopy longwave radiation on snow cover and/or snowmelt had not been part of our initial submission. We included comparison of meltout between simulations and a state-of-the-art snow water equivalent product in this revision, which revealed a general delay of snow-off dates across boreal forests in simulations. Correction of sub-canopy longwave radiation was found to decrease this bias.**

*A blended data set of five global observation-based SWE products (henceforth, Blended-5) covering the period 1981 to 2010 (Mudryk et al., 2015) was used to estimate snow-off dates across the Northern Hemisphere and evaluate simulation of snowmelt in CTRL and CORR. In contrast to simulations, observations display snow persisting for physically unrealistical durations, which necessitates a SWE threshold to estimate snow-off dates (Krinner et al., 2018). While Mudryk et al. (2017) and Krinner et al. (2018) used thresholds of 4mm and 5mm, respectively, for estimates of spatial snow cover extent, a smaller SWE value was necessary to represent the precise timing of meltout within individual grid cells. A threshold of 1mm was used in this study to define meltout for the Blended-5 mean, and snow-off date was defined as the first day of a year for which SWE did not exceed this threshold. Sensitivity of snow-off dates to threshold values was tested for the range 0.5mm to 4mm, however, the overall conclusions of this study are unchanged for different thresholds.*

*Simulated and observed snow-off dates are compared in Fig. 10 for grid cells with consistent snow cover throughout preceding December and coverage by evergreen needleleaf trees of at least 50%. Simulations CTRL and CORR generally feature a narrower probability density function (PDF) of snow-off dates, indicating a shorter snowmelt season, and later meltout compared to observations across the entire Northern Hemisphere (Fig. 10a). While shapes of observed PDFs are well represented by*

*simulations over Eurasia (Fig. 10b, d), observations show a clearer, shorter peak of meltout compared to simulations over mountainous western North America (Fig. 10c). Correction of sub-canopy longwave radiation displays little impact when accumulated over the entire Northern Hemisphere, however, it systematically reduces the delay of simulated snow-off dates throughout the snowmelt season. PDFs of snow-off dates for regional subsets reflect spatial patterns seen in Fig. 7h, with minor differences between CTRL and CORR over most of western North America (Fig. 10c) and eastern Siberia (Fig. 10d) but substantial acceleration of snow-off dates over western Siberia and eastern Europe (Fig. 10b) due to correction of sub-canopy longwave radiation.*

*The regionally limited impact of corrected sub-canopy longwave radiation is highlighted by filtering PDFs of snow-off date for grid cells with average differences in snow-off date between CORR and CTRL of at least 3 days (Fig. 10e, f). Correction of sub-canopy longwave radiation improves timing of meltout in filtered grid cells, especially over western Siberia and eastern Europe where the filtered PDF for CORR, in contrast to CTRL, closely resembles observations. PDFs of snow-off dates derived from Blended-5 SWE display sensitivity to threshold choices, however, this uncertainty is generally smaller than differences between simulations and observations.*

For the near future for validation and/or assimilation, the authors could mentioned the new radar Sentinel datasets that allow to monitor the wet snow evolution through open forest canopy (Small, David; Rohner, Christoph; Miranda, Nuno; Rüetschi, Marius; Waser, Lars; Vögtli, Marius; Schaepman, Michael E (2018). Level 3 wide-area backscatter time-series for wet-snow mapping and forest classification. In: EGU General Assembly 2018, Vienna, 8 April 2018 - 13 April 2018.). For closed forest area, the uncertainties seem still important?

**Thank you very much for this suggestion! We did not know about this recent development.**

Please clarify how you estimate the "Longwave enhancement" parameter?

**Values for longwave enhancement are calculated as the ratio of below-canopy to above-canopy longwave radiation. As this is stated in the introduction, we did not take any action.**

Why do you differentiated day to night sky emissivities? (Fig. 3 d and e)

**We initially tried to create a correction for both daytime and nighttime sub-canopy longwave radiation, but found that one set of regression coefficients led to unrealistic variations in sub-canopy longwave radiation around noon as well as to inconsistencies in sub-canopy longwave radiation at sunset and (mainly) sunrise. Calculation of separate regression coefficients for day and night resulted in smoother transitions of sub-canopy longwave radiation between day and night, and one potential reason for this is the impact of topography during low solar elevation, especially at (sub-)alpine sites Alptal and Seehornwald. Note that daytime regression coefficients would yield different correction factors for zero insolation than nighttime regression coefficients do, which might be due to other variables also governing deficiencies in simulated sub-canopy longwave radiation as our corection only explains 60% of variance in simulation errors. However, correction reducing diurnal ranges in sub-canopy longwave radiation does not depend on separating daytime and nighttime regression coefficients.**

Which ground (or snow?) emissivity value do you consider?

**CLM4.5 calculates ground emissivity as a weighted sum of emissivities of snow (0.97) and soil (0.96), weighted by the fraction of snow covering the grid cell.**

[Figure]

**This has been added to the description of equation (4).**

*...with emissivity of the ground $\varepsilon_g$ and temperature of the ground $T_g$. Ground emissivity in CLM4.5 is calculated as a weighted sum of emissivities of snow (0.97) and soil (0.96), weighted by the fraction of snow covering a grid cell.*

---

## Author Comment (AC2) · 20 Jun 2019

This document contains comments by reviewers (regular font), **responses to reviewers (bold)**, and *citations from the manuscript (italic)* including additions to the manuscript (blue).

Reviewer 2 Comments:

Technically, the demonstration is not thoroughly made, that single-layer canopy formulations generate melt delay in the NH with respect to "real world" (that here would be observations). Indeed, the demonstration of this effect is only made with respect to an approximation of a 2-layer model, that may itself be heavily biased. This is all the more worrying as Todt et al., 2018, illustrate an increase in a subcanopy LW positive bias upon the use of a 2-layer model at the Seehornwald conifer site (Todt et al., 2018, Figure 5).

**While the two-layer canopy model SNOWPACK did show a positive bias at the Seehornwald site, it also showed a substantial negative bias at the site of So-dankylä. These biases stem from SNOWPACK being calibrated for the site of Alptal, which features a lower vegetation density than Seehornwald but a larger vegetation density than Sodankylä. Consistent for those three sites was the substantially smaller spread in and diurnal cycle of sub-canopy longwave radiation shown by Todt et al. (2018). Furthermore, Gouttevin et al. (2015) showed that the improvement from one layer to two layers did result in a reduction of the negative bias that SNOWPACK displayed for Alptal, although this did not (just) originate from longer nights than days but (also) from larger nighttime underestimations than daytime overestimations. But it is true that the delay in meltout found between global simulations had not been shown relative to observations, instead had been inferred as a consequence from comparison of sub-canopy longwave radiation with observations at forest-stand scales. Evaluation of global simulations has been added as described in the next paragraph.**

The melt delay associated with 1-layer canopy models claimed by the authors, may clearly be real, but the demonstration should be improved, for instance by confrontation of simulation results to (i) in-situ data at field sites and (ii) satellite estimate of NHsnow disappearance dates. Comparison of simulation results to observed snow-cover fractions is briefly mentioned in the Discussion while it should be an important

part of the Result section (as it is already quite well advertised as a rationale for this study in the Introduction). Thackeray et al., 2015 (their Figure 3 for instance) provides a good baseline for such comparisons.

**We do understand and acknowledge the value of a comparison to observed snow-off dates, and we have included comparison of meltout between simulations and a state-of-the-art snow water equivalent product in this revision. This comparison revealed a general delay of snow-off dates across boreal forests in simulations, and correction of sub-canopy longwave radiation was found to decrease this bias. However, comparison at Toy Model sites is challenging and potentially inconclusive as snow measurements are largely unavailable. Various approximations would have to be used for comparison as was done for driving the Toy Model by Todt et al. (2018), and the use of this might be limited. Therefore, we restricted evaluation to global snowmelt.**

*A blended data set of five global observation-based SWE products (henceforth, Blended-5) covering the period 1981 to 2010 (Mudryk et al., 2015) was used to estimate snow-off dates across the Northern Hemisphere and evaluate simulation of snowmelt in CTRL and CORR. In contrast to simulations, observations display snow persisting for physically unrealistical durations, which necessitates a SWE threshold to estimate snow-off dates (Krinner et al., 2018). While Mudryk et al. (2017) and Krinner et al. (2018) used thresholds of 4mm and 5mm, respectively, for estimates of spatial snow cover extent, a smaller SWE value was necessary to represent the precise timing of meltout within individual grid cells. A threshold of 1mm was used in this study to define meltout for the Blended-5 mean, and snow-off date was defined as the first day of a year for which SWE did not exceed this threshold. Sensitivity of snow-off dates to threshold values was tested for the range 0.5mm to 4mm, however, the overall conclusions of this study are unchanged for different thresholds.*

*Simulated and observed snow-off dates are compared in Fig. 10 for grid cells with*

*consistent snow cover throughout preceding December and coverage by evergreen needleleaf trees of at least 50%. Simulations CTRL and CORR generally feature a narrower probability density function (PDF) of snow-off dates, indicating a shorter snowmelt season, and later meltout compared to observations across the entire Northern Hemisphere (Fig. 10a). While shapes of observed PDFs are well represented by simulations over Eurasia (Fig. 10b, d), observations show a clearer, shorter peak of meltout compared to simulations over mountainous western North America (Fig. 10c). Correction of sub-canopy longwave radiation displays little impact when accumulated over the entire Northern Hemisphere, however, it systematically reduces the delay of simulated snow-off dates throughout the snowmelt season. PDFs of snow-off dates for regional subsets reflect spatial patterns seen in Fig. 7h, with minor differences between CTRL and CORR over most of western North America (Fig. 10c) and eastern Siberia (Fig. 10d) but substantial acceleration of snow-off dates over western Siberia and eastern Europe (Fig. 10b) due to correction of sub-canopy longwave radiation.*

*The regionally limited impact of corrected sub-canopy longwave radiation is highlighted by filtering PDFs of snow-off date for grid cells with average differences in snow-off date between CORR and CTRL of at least 3 days (Fig. 10e, f). Correction of sub-canopy longwave radiation improves timing of meltout in filtered grid cells, especially over western Siberia and eastern Europe where the filtered PDF for CORR, in contrast to CTRL, closely resembles observations. PDFs of snow-off dates derived from Blended-5 SWE display sensitivity to threshold choices, however, this uncertainty is generally smaller than differences between simulations and observations.*

Secondly, for the evaluation of the delay effect against in-situ data, simulation errors coming from the meteorological forcing should be minimized. Evergreen forest sites used in Todt et al., 2018, provide appropriate observed meteorological data. They could be used in the place of erroneous large-scale meteorological forcing data (for e.g. Figure 4 and the associated results analysis).

As mentioned in the previous comment, snow cover measurements at Toy Model sites are either available only by approximation or not at all. Replicating Figure 4 and its analysis with stand-scale simulations is possible, and the impact of correction on sub-canopy longwave radiation indeed leads to improved simulations. However, this should be expected as observations of sub-canopy longwave radiation from these forest stands are used to create the correction. Therefore, we test and train correction of sub-canopy longwave radiation with the same dataset, and this comparison on forest-stand scales is a further illustration rather than an evaluation, which is why we decided against its inclusion in our manuscript.

p1L10 : the last sentence of the abstract associates "boreal forests" and "warm winters" where "snowmelt occurs early". This is not very intuitive

**"Warm winters" are meant to be relative to regions where there is snow. The chain of causation is warmer winters → earlier snowmelt → snowmelt when nights are longer than days → substantial underestimation of sub-canopy longwave radiation → delay of snowmelt. We deleted the part about warmer winters as early snowmelt already indicates that these regions are warmer than other snow-covered regions.**

*Increasing insolation and day length change the impact of overestimated diurnal cycles on daily average sub-canopy longwave radiation throughout the snowmelt season. Consequently, delay of snowmelt in land-only simulations is more substantial where snowmelt occurs early.*

p2L3 : cite Krinner et al., 2018

**We thank the reviewer for pointing out this paper. However, we feel that is does not relate closely to the material being discussed in that section because, while ESM-SnowMIP will include an experiment to investigate the impact of vegetation distribution, the rationale of Krinner et al. (2018) and its references are focused on surface albedo. Also, modification of vegetation distribution will not have an impact on the control of vegetation density on vegetation temperatures. We therefore elect to retain the original text.**

p3L9-12 : "emissivity" is unappropriately used. What the authors call "emissivity", is an unexplained combination of emissivity and sky view factor. Please explicit and justify the approximations that you make here.

**Yes, the authors absolutely agree that the parameter "vegetation emissivity" used by CLM4.5 is not an emissivity in the physical sense, i.e. as an emissivity is commonly used in the Stefan-Boltzmann equation. However, this is the nomenclature used by CLM4.5, which we decided to stick to in order for consistency with CLM4.5, and the technical description of CLM4.5 (Oleson et al., 2013) does not give any reasoning for or description of the combination of actual physical emissivity and SVF/canopy coverage.**

p3L9-12 : please explain briefly how Tv is calculated.

**A short description of the calculation of Tv by CLM4.5 has been added.**

*...using the Stefan-Boltzmann law with Stefan-Boltzmann constant $\sigma$ = 5.67 $10^{-8}\ Wm^{-2}K^{-4}$ and vegetation temperature $T_v$. Vegetation temperature is calculated based on an energy balance, net radiation minus turbulent heat fluxes. Radiative trans-*

*fer of direct and diffuse shortwave radiation is calculated via a two-stream approximation (Sellers, 1985) considering one reflection from ground to canopy. Net longwave radiation is calculated from atmospheric longwave radiation, vegetation temperature, and (ground) surface temperature and determined by vegetation emissivity and emissivity of the ground. Calculation of turbulent heat fluxes in CLM4.5 is based on Monin-Obukhov similarity theory and described by Oleson et al. (2013). Vegetation emissivity depends on Leaf Area Index ($LAI$) and Stem Area Index ($SAI$) and is calculated as...*

P6L11-13 : In the current structure of the paper, calculating and exhibiting correction factors for deciduous forest regions makes in my opinion little sense, as no use is made of them, and very little analysis of their difference w/r to coefficients calculated for evergreen forests is made. If Cherskii's coefficients are similar to those from evergreen sites, then what is the added value of including this site in the calibration intended for evergreen sites ?! To me, it just undermines the calibration approach.Calculating correction coefficients for deciduous forests stands may be interesting, though, providing their difference (w/r evergreen sites) and impact (on e.g. snowmelt) is discussed in the paper.

**As this paper is a continuation of Todt et al. (2018), we included all sites from that paper that could be used for multiple linear regression. The reasons why sub-canopy longwave radiation is only corrected for evergreen trees and why Cherskiy is included in calculating this correction are explained in the paper. Simulations for Cherskiy are similar to evergreen trees while Abisko is not reliable/sufficient. This similarity is then used to balance very dense (Alptal, Seehornwald) and sparser sites (Cherskiy, Sodankylä) for a more general applicability. Similarities and differences between regression coefficients for Abisko, Cherskiy, and evergreen forest stands are indeed interesting and open up a separate discussion about the influence of stand characteristics and structure on**

sub-canopy radiative fluxes. For the sake of clarity and to keep the focus of our paper on the impact of corrected sub-canopy longwave radiation, we (had) decided against including that discussion.

p6L20 : please add 'CLM4.5' before 'grid cell' for more clarity

**Has been added. Thank you for this suggestion.**

*For the location of Alptal, in contrast to other forest stands used in this study, forest stand and CLM4.5 grid cell feature similarly high vegetation densities (PAIs of 4.1 $m^2$ $m^{-2}$ and 3.7 $m^2$ $m^{-2}$, respectively) and thus similar vegetation emissivities $\varepsilon_v$ (0.983 and 0.975, respectively). This allows for a comparison of diurnal cycles of sub-canopy longwave radiation as well as longwave enhancement between stand-scale measurements and offline simulations.*

P6paragraph4.1 : I suggest to carry this analysis using an observed forcing to evidence the effect of going from 1-layer to 2-layer w/r to observations in an "ideal" case. Also, it should be mentionned more clearly in the text that the selected period corresponds to the snowmelt season at Alptal and hence is relevant to assess the effect of subcanopy LW on snowmelt.

**This analysis is done to highlight the (successful) effect on diurnal cycles of sub-canopy longwave radiation rather than as an evaluation. The Toy Model also does not include all calculations/parameterizations of CLM4.5, only those necessary to simulate sub-canopy longwave radiation while offline simulations model the entire land surface, so the impact on sub-canopy longwave radiation simulated by the Toy Model does not tell the entire story. Caption of Figure 4**

**mentions that the snowmelt season at Alptal is shown, and this is added to the text.**

*Implementation of correction factors in CLM4.5 results in decreased sub-canopy long-wave radiation during daytime and increased sub-canopy longwave radiation during nighttime, thereby reducing diurnal cycles. For the grid cell representing Alptal, diurnal ranges decrease from about 70 W m$^{-2}$ to about 30 W m$^{-2}$ during snowmelt season (Fig. 4a and Fig. 4b). Observations at the forest stand show an average diurnal range of about 15 W m$^{-2}$ during snowmelt season.*

P7Paragraph4.2 : Maybe other forcings than CRUNCEP, exhibit less bias with respect to spatio-temporal in sky emissivity. Brun et al., 2013, concluded that ERA-i generally leads to improved simulations (w/r to other forcings) over large areas of the N high latitudes.

**Only two datasets are options in CLM4.5 for forcing of offline simulations – the CRUNCEP dataset used in this study and a dataset described by Qian et al. (2006). Snow cover in CLM4.5 offline simulations driven with both of these datasets have been analyzed by Thackeray et al. (2015), revealing higher skill scores for the simulation driven by the CRUNCEP dataset. Because of this and because of shorter coverage by the Qian dataset, we decided to use and stick to CRUNCEP forcing data.**

**References:**

**Qian, T., A. Dai, K. E. Trenberth, and K. W. Oleson (2006), Simulation of global land surface conditions from 1948 to 2004. Part I: Forcing data and evaluations, J. Hydrometeorol., 7(5), 953–975, doi:10.1175/JHM540.1.**

**Thackeray, C.W., C. G. Fletcher, and C. Derksen (2015), Quantifying the**

skill of CMIP5 models in simulating seasonal albedo and snow cover evolution, Journal of Geophysical Research: Atmospheres, 120, 5831–5849, doi:10.1002/2015JD023325.

P8L3 : I suggest to outline the regions wth frac_PFT>0.5 on these maps, for better understanding of the effects of CORR and their magnitude.

**A contour line for fractional coverage by evergreen needleelaf trees of at least 50% has been added. Thank you for this suggestion.**

P8L22:maybe the glaciated areas should be masked out as they are not the focus of this study. Otherwise, the question arises as to whether cold content for these areas refers to the snow, or to thewhole snow+ice columns.

**The reviewer is correct that cold content in glaciated regions includes ice, which is why explicit values for these regions are not mentioned in the text and the colorbar is cropped at 5 MJ m$^{-2}$. The choice to include these grid cells was made for visual reasons, as many grid cells that include the landunit "glaciated" are entirely glaciated and would thus appear as empty on maps. Therefore, we decided to take no action.**

P8L30 : this is not true to my understanding as melt out date is largely determined by the energy required to melt the snow (which is often higher than the one needed to raise snow temperature to 0°C)

**We agree with the reviewer and the statement made at that point is not intended**

[Figure]

**to suggest that spatial differences in cold content solely cause the spatial pattern in meltout. However, there is a clear spatial correlation between these two variables, especially between changes in meltout and realtive changes in cold content, and more and/or colder snow inevitably requires a higher energy input for melt. Which is why we decided to take no action.**

P9L1 : maybe the delay between melt-out-date, and equinox, could be an interesting additionnal explanatory variable in Fig8. Also, an illustation of daily cycle changes before and after equinox for Southeastern regions would be a great complement to the explanations.

**The difference between equinox and melt-out date is indeed one of the major governing variables of the impact of corrected sub-canopy longwave radiation. However, grid cells across Siberia generally feature meltout past the equinox, and location (latitude/insolation, elevation) appears to be the most important characteristic for this region, which is why we decided not to add a panel for difference to equinox. Difference to equinox is a more helpful metric when assessing the impact on snowmelt across regions with starker contrasts in meltout date, e.g. Europe**.

**The illustration of changes in diurnal cycles only has limited explanatory value, as daytime overestimations and nighttime underestimations are fairly consitent and changes in day length are not overly striking when visualized as a diurnal cycle. In fact, Figure 4b already features diurnal cycles of sub-canopy longwave radiation prior to the boreal spring equinox and post-equinox diurnal cycles would only appear slightly different. However, we included a figure showing Northern Hemisphere maps of changes in longwave enhancement due to correction before and after the equinox (see last comment in this review).**
P9L15 : specify "over the study region"

**The particular statement holds generally and should not be limited to our study region, as the governing factors are day length, snow cover, and presence of evergreen needleleaf trees. Therefore, we decided to stick to the original phrasing.**

P9L30 : Illustrations of comparisons to observations like in Thackeray et al. 2014, 2015, is exactly what is missing in your study, and would add great value to it.

**We have included comparison of meltout between simulations and a state-of-the-art snow water equivalent product in this revision, which revealed a general delay of snow-off dates across boreal forests in simulations. Correction of sub-canopy longwave radiation was found to decrease this bias. Detailed additions to the manuscript are listed underneath the first comment of this review.**

Overall, there is a lack of proper illustration of the competing effects of CORR on the daily subcanopy LW radiation before vs after the equinox, and how this governs the effects of CORR at the global scale.

**We added a figure showing changes in longwave enhancement by evergreen needleleaf trees across the Northern Hemisphere before and after the boreal spirng equinox, which show contrasting signs in the impact of corrected subcanopy longwave radiation.**

*As offline simulations lack spatial variability in $\varepsilon_{sky}$, latitude (through insolation) and duration of snow on the ground (through day length) control spatial differences in impact of correction of sub-canopy longwave radiation on snow-off date. Changes in long-*

*wave enhancement due to correction of sub-canopy longwave radiation before and after the boreal spring equinox, approximated by averages over February/March and April/May, display opposite signs across the Northern Hemisphere (Fig. 8), with shorter (longer) days than nights before (after) the equinox resulting in an increase (decrease) in daily average longwave enhancement. Generally, lower insolation at higher latitudes leads to a more positive impact of correction on daily average longwave enhancement, increasing (decreasing) positive (negative) changes in longwave enhancement with increasing latitude before (after) the boreal spring equinox. Across mid-latitudes, increase in daily average longwave enhancement over February and March is roughly similar to decrease in daily average longwave enhancement over April and May, while increase over February and March outweighs decrease over April and May across high latitudes including most of the regions covered by boreal forests.*

---

## Author Response (AR1)

**Simulated single-layer forest canopies delay Northern Hemisphere snowmelt**

[revised manuscript text omitted]

Conceptually, correction factors represent a vegetation (Added:) structure consisting of multiple individual layers, so that longwave radiation fluxes emitted upward and downward from the vegetation are no longer equal by design. In a multi-layer canopy (Deleted:)  (Added:) configuration, the uppermost layer contributes most to longwave radiation emitted upward (Deleted:)  (Added:) to the atmosphere and directly absorbs incoming longwave and shortwave radiation fluxes. Conversely, the lowest layer contributes most to longwave radiation emitted downward (Deleted:)  (Added:) to the surface, but is insulated from atmospheric fluxes by the canopy layers above.

Using (Added:) this multi-layer canopy (Deleted:)  (Added:) configuration as a guideline, longwave radiation (Added:) emitted by the canopy was (Deleted:)  (Added:) partitioned asymmetrically upward and downward in CLM4.5 (Deleted:)  (Added:) . The resulting above-canopy longwave radiation (Added:) 
[revised manuscript text omitted]

---

## Referee Report (RR1)

I thank the authors for the interesting discussion and reactions they provided.

All comments have been thoroughly adressed and I am satsified with the modifications performed to the manuscript.

I however wish to « negociate » with the authors some very minor points as a kind of « trade-off » between original remarks from my side, and their answer :

1- about the use of « emissivity » : as the use of this word here in just physically unproper, but considering the reasons why the authors stand to it, I would suggest the following changes in the manuscript (in blue) :

« Vegetation emissivity depends on Leaf Area Index (LAI) and Stem Area Index (SAI) and is calculated as :

$\varepsilon_v = 1 - e^{-(LAI+SAI)}$ .

This parameter is not an emissivity in the physical sense but we stick to this denomination here for consistency with the nomenclature of the technical description of CLM4.5 (Oleson et al., 2013). »

2- regarding cold content : I still think that the sentence P10 L1 « Spatial patterns in snow-off date are similar to those in cold content as higher cold content corresponds to later meltout »  is misleading as the use of « as » suggests a causal relationship which is not universal (cf the role of SWE on the required melt energy, disregarding the value of the cold content). I therefore suggest the following change :

« Spatial patterns in snow-off date are similar to those in cold content with higher cold content corresponding to later meltout »